# Fucoidan Protects against Doxorubicin-Induced Cardiotoxicity by Reducing Oxidative Stress and Preventing Mitochondrial Function Injury

**DOI:** 10.3390/ijms231810685

**Published:** 2022-09-14

**Authors:** Yuting Ji, Dekui Jin, Jingyi Qi, Xuan Wang, Chengying Zhang, Peng An, Yongting Luo, Junjie Luo

**Affiliations:** 1Department of Nutrition and Health, Beijing Advanced Innovation Center for Food Nutrition and Human Health, China Agricultural University, Beijing 100193, China; 2Department of General Practice, The Third Medical Center of Chinese PLA General Hospital, Beijing 100039, China

**Keywords:** fucoidan, doxorubicin, cardiotoxicity, oxidative stress, mitochondrial function

## Abstract

Doxorubicin (DOXO) is a potent chemotherapeutic drug widely used to treat various cancers. However, its clinical application is limited due to serious adverse effects on dose-dependent cardiotoxicity. Although the underlying mechanism has not been fully clarified, DOXO-induced cardiotoxicity has been mainly attributed to the accumulation of reactive oxygen species (ROS) in cardiomyocytes. Fucoidan, as a kind of sulphated polysaccharide existing in numerous brown seaweed, has potent anti-oxidant, immune-regulatory, anti-tumor, anti-coagulate and anti-viral activities. Here, we explore the potential protective role and mechanism of fucoidan in DOXO-induced cardiotoxicity in mice. Our results show that oral fucoidan supplement exerts potent protective effects against DOXO-induced cardiotoxicity by reducing oxidative stress and preventing mitochondrial function injury. The improved effect of fucoidan on DOXO-induced cardiotoxicity was evaluated by echocardiography, cardiac myocytes size and cardiac fibrosis analysis, and the expression of genes related to cardiac dysfunction and remodeling. Fucoidan reduced the ROS content and the MDA levels but enhanced the activity of antioxidant enzymes GSH-PX and SOD in the mouse serum in a DOXO-induced cardiotoxicity model. In addition, fucoidan also increased the ATP production capacity and restored the levels of a mitochondrial respiratory chain complex in heart tissue. Collectively, this study highlights fucoidan as a potential polysaccharide for protecting against DOXO-induced cardiovascular diseases.

## 1. Introduction

Heart diseases and cancer are the main causes of mortality and morbidity in industrial countries, in particular, cancer remains the most common cause of death around the world [1,2]. Chemotherapeutics have been confirmed to effectively improve survival rates in cancer, while they generate a wide range of cardiovascular complications, especially cardiotoxicity [3]. Doxorubicin (DOXO) is a broad-spectrum anthracycline antibiotic drug, and DOXO-based chemotherapy remains the cornerstone in cancer treatment, which is used for a variety of cancers, such as breast cancer, solid tumors, leukemia and soft-tissue sarcomas [4,5]. However, the therapeutic values and clinical applications of DOXO are limited by the life-threatening cardiotoxicity, both acute and chronic, including arrhythmia, tachycardic and even serious congestive heart failure and left ventricular dysfunction [6,7,8]. Indeed, DOXO often causes cell atrophy, while DOXO induction increases cardiac volume and causes cardiac hypertrophy [9,10]. In a study of 4018 patients with cancer treated with DOXO, DOXO-induced cardiotoxicity was severe, with 2.2% experiencing symptoms of heart failure [11]. Additionally, DOXO-induced cardiotoxicity is always intensified in patients with an increased dose of DOXO [12]. The mechanisms of DOXO-induced cardiotoxicity appear to be multiple, involving oxidative stress, lipid peroxidation, mitochondrial injury, DNA/RNA damage, autophagy and apoptosis [13,14]. However, oxidative stress is considered a key process in DOXO-induced myocardial damage. DOXO can induce a large amount of reactive oxygen species (ROS), which in turn causes mitochondrial dysfunction and cellular injury [5,15]. Hence, reducing oxidative stress may be an effective way to treat and prevent DOXO-induced cardiotoxicity.

Fucoidan is a natural active sulfate polysaccharide mainly existing in the cytoplasmic matrix of brown algae [16], which has attracted extensive attention due to its various biological activities [17]. Fucoidan has been reported to play a role in anticoagulation and antithrombosis [18]; anti-tumor activities [19]; antiviral and antioxidation activities, and anti-inflammation [20], and it also has effects on lipid reduction, increased kidney [21] and liver activity [22], and gastrointestinal protection [23,24]. In recent years, numerous studies have indicated that fucoidan possesses strong antioxidant activities in vivo and in vitro [25,26]. In addition, fucoidan has been shown to participate in improving mitochondrial function and attenuating oxidant stress [27]. Evidence has demonstrated that fucoidan could improve cardiac metabolism and function in aging mice [28], which shows its potential to protect against cardiac impairment.

The heart is the largest energy-consuming organ in the human body, and one of the cells with the most amount of mitochondria is cardiomyocytes. Almost 95% of ATP consumed by the heart comes from the oxidative metabolism of mitochondria [29]. Notably, evidence showed that mitochondrial dysfunction in mouse hearts can induce progressive cardiac hypertrophy with systolic dysfunction [30]. DOXO has been reported to impair cardiac function mainly through oxidative stress and mitochondrial dysfunction [31]. Therefore, strategies for mitigating oxidative stress and enhancing mitochondrial function are valuable and can potentially be employed to inhibit the progression and development of DOXO-induced cardiotoxicity. The objective of this study is to assess whether DOXO-induced cardiotoxicity in mice can be prevented by exogenous administration of fucoidan by decreasing oxidative stress and protecting mitochondrial function.

## 2. Results

### 2.1. Fucoidan Protects against DOXO-Induced Left Ventricular Dysfunction

We designed experiments to detect the protective effect of fucoidan on DOXO-induced heart damage in mice (Figure 1A). One group of mice was fed with just water for 21 days (control group, ctl). The second group of mice was fed with water for 14 days prior to a cumulative dose of 7 mg/kg DOXO for seven daily intraperitoneal injections (DOXO group). The mice of group three were first treated with fucoidan for 14 days, and DOXO and fucoidan were then administered together for the following 7 days (fucoidan + DOXO group).

To verify whether fucoidan could protect against DOXO-induced cardiotoxicity, we tested some index about the transthoracic echocardiogram in vivo. The results show that the LV end-systole internal dimension (LVID) is 2.456 ± 0.081 mm in the DOXO group vs. 1.432 ± 0.043 mm in the ctl group (*p* < 0.001), that the LV end-diastolic internal dimension (LVIDd) is 3.882 ± 0.022 mm in the DOXO group vs. 3.197 ± 0.076 mm in the ctl group (*p* < 0.001), that the LV end-systole volume (LVESV) is 16.279 ± 0.660 mm^3^ in the DOXO group vs. 9.177 ± 0.817 mm^3^ in the ctl group (*p* < 0.001) and that the LV end-diastolic volume (LVEDV) is 63.031 ± 1.184 mm^3^ in the DOXO group vs. 41.128 ± 2.323 mm^3^ in the ctl group (*p* < 0.001) (Figure 1B,C; Table 1). Therefore, the left ventricular function of the mice was significantly damaged after DOXO injection compared with the ctl group. Nevertheless, the DOXO injection-induced cardiac injury can be improved by fucoidan gavage. The mice detection of the fucoidan + DOXO group shows that LVIDs is 1.210 ± 0.077 mm, that LVIDd is 3.051 ± 0.035 mm, that LVESV is 6.716 ± 0.934 mm^3^ and that LVEDV is 36.582 ± 0.996 mm^3^ (Figure 1B,C; Table 1), all of which returned to the levels of the ctl group, suggesting the protective effect of fucoidan.

### 2.2. Fucoidan Attenuates DOXO-Induced Myocardial Atrophy and Cardiac Fibrosis

Two important markers of DOXO-induced cardiotoxicity, cardiomyocyte size and cardiac fibrosis, were analyzed by H&E staining and PicroSirius red staining. The results of H&E staining showed that myocardial cells in the DOXO group were smaller and that interstitial cracks were larger than those in the ctl group, while these lesions were significantly ameliorated in mice given fucoidan gavage (Figure 2A). In addition, PicroSirius red staining showed that DOXO injection increased the rate of interstitial fibrosis, and fucoidan can prevent this injury significantly (Figure 2B).

### 2.3. Fucoidan Prevents DOXO-Induced Cardiac Dysfunction and Structure Damage

By measuring serum levels of enzymes that are hallmarks of cardiac function, the contents of AST, CK and its isoenzyme CK-MB, and LDH and its isoenzymes LDH-1 in the mice of DOXO group were significantly higher than those in the ctl group (Figure 3A,B), while fucoidan was capable of lowering their expression levels, and the expression levels of these markers were nearly the same as in the ctl group, which suggests that fucoidan can reduced DOXO-induced myocardial damage (Figure 3A,B). Furthermore, the mRNA expression levels of hallmark genes of cardiac dysfunction were also detected among the three groups. The mRNA expression of atrial natriuretic peptide (*ANP*), brain natriuretic peptide (*BNP*) and Myosin Heavy Chain 7 (*Myh7*) increased in the mice of the DOXO group, but these genes expression decreased in the mice pre-treated with fucoidan compared with those in the DOXO group (Figure 3C). Meanwhile, the expression of connective tissue growth factor (*CTGF*) and matrix metalloproteinase-2 (*MMP-2*) significantly increased after DOXO injection, which was consistent with the increases in *ANP* and *BNP*, and the combination treatment with fucoidan can alleviate these increases (Figure 3D). Therefore, the results of our experiments showed that oral fucoidan administration has a protective effect on heart function.

### 2.4. Fucoidan Blunted the DOXO-Induced Increase in Oxidative Stress in Serum and Cardiac Tissue

Excessive production of ROS in cells is a key indicator of oxidative stress, and the level of ROS was detected in every mouse cardiac tissue of each group (Figure 4A). As described in the figure, mice in the DOXO group had a higher level of ROS than those in the ctl group, while fucoidan administration improved this situation. Similarly, the levels of MDA (the most universal by-product of lipid peroxidation) in the serum of the mice indicated that DOXO induced a higher risk of lipid peroxidation, and this process was inhibited by fucoidan (Figure 4B). Meanwhile, we measured the activity of antioxidant enzymes GSH-PX and SOD in the serum and found these enzyme activities in DOXO injected mice was significantly reduced while fucoidan administration prevented the activity decrease and improved the oxidative injury (Figure 4C,D).

### 2.5. Fucoidan Attenuates DOXO-Induced Mitochondrial Dysfunction in Cardiac Tissue

Mitochondria play a key role in determining cardiomyocyte viability [32]. It is necessary to detect mitochondrial dysfunction in DOXO-induced cardiotoxicity. ROS was considered by-products in the process of ATP production through the respiratory chain [33], and we have detected a significant increase in ROS in DOXO-induced myocardial injury. Then, we tested the levels of ATP in mouse cardiac tissue of all three groups and showed that the ATP level was lower of in the DOXO group than in the ctl group but restored in the fucoidan + DOXO group (Figure 5A).

The ability of mitochondria to produce ATP is closely related to the gene expression of the mitochondrial oxidative respiratory chain. We therefore examined the expression of mitochondrial DNA-encoded mitochondrial cytochrome b (*mt-Cytb*) and mitochondrial ATPase6 (*mt-ATP6*) (Figure 5B). The mRNA expression levels of these two genes were downregulated in the DOXO group but recovered in the fucoidan + DOXO group. In addition, we measured the marker genes (*NDUFB8*, *SDHB*, *UQCR2*, *MTCO2* and *ATP5F1*)-related mRNA expression of mitochondrial respiratory chain complex І–V in the heart tissues of each mouse (Figure 5C). The expression levels of these genes in the DOXO-injected group were significantly lower than those in the ctl group, indicating that DOXO probably disrupted the function of mitochondrial respiratory chain complexes, ultimately leading to reduced ATP production. Not surprisingly, fucoidan can significantly improve these processes. To sum up, our results suggested that DOXO-induced damage of cardiac mitochondrial function can be prevented by fucoidan.

## 3. Discussion

Abundant evidence has revealed that DOXO could lead to acute or/and chronic cardiotoxicity. The risk of heart failure was 5% on average at a cumulative dose of 400 mg/m^2^, but it was increasing exponentially at higher doses [12,34]. The incidence of congestive heart failure exceeded 26% at a cumulative dose of 550 mg/m^2^ [35] and increased to 48%, with a cumulative dose of 700 mg/m^2^ [36]. The related mechanisms were multiple, including autophagy dysregulation, inhibition of DNA/RNA/protein synthesis and intracellular Ca^2+^ dyshomeostasis [37,38]. However, emerging studies have proposed the indispensable role of the excessive production of ROS in DOXO-induced myocardial injury [39]. DOXO induces abnormally high levels of oxygen free radicals in the heart, which can damage myocardial function and even lead to heart failure [40,41]. Therefore, efficient strategies are needed to reduce the oxidant stress caused by DOXO.

There are current efforts on the treatment of DOXO-induced cardiotoxicity, such as iron chelator drugs [42] and free-radical scavengers. However, given that these treatments are limited by a variety of side effects, it is necessary to find safe and effective substances to protect against heart damage caused by DOXO. Natural substances play a significant role in drug discovery, and researchers have struggled to find some effective natural antioxidant products to improve DOXO-induced cardiotoxicity. Here, we found a marine substance, fucoidan, which is derived from brown algae and consists of sulfate groups and fucose, has been extensively studied in recent years due to its good pharmacological effects and many biological activities.

In this study, our experimental design included the control group, the DOXO group and the fucoidan + DOXO group, which is reasonable and sufficient to demonstrate the protective effect of fucoidan against doxorubicin-induced cardiotoxicity. We believe that the control group treated with fucoidan alone is unnecessary for the following reasons. (1) The control group treated with fucoidan alone may be used to detect potential toxic side effects of fucoidan, but it is currently believed that fucoidan is a natural polysaccharide derived from food, and the safety and availability of fucoidan as a food supplement or dietary supplement have been established and verified [43,44]. (2) More importantly, the toxic side effects of fucoidan are not the focus of this paper. (3) Consistent with our study design, many similar published articles used the same grouping strategy in animal experiments [45,46,47,48]. Although a previous study showed that fucoidan from fucus vesiculosus can attenuate DOXO-induced acute cardiotoxicity by regulating apoptosis and autophagy in rat [49], more mechanisms remain to be explored. In fact, cardiac dysfunction and injury can be protected by maintaining the mitochondrial function and mitochondrial mass [50,51]. Moreover, an enhancement of the mitochondrial biogenesis and capability also has protective effects against cardiac functional damage [52]. In the present study, we observed that mice intraperitoneal injection with DOXO had abnormalities in cardiac pumping function and cardiac structure, which were significantly ameliorated by the oral administration of fucoidan. In addition, fucoidan improved DOXO-induced cardiomyocyte shrinkage and cardiac fibrosis. Mechanistically, DOXO administration resulted in left ventricular and mitochondrial dysfunction and increased oxidative stress, while these negative effects were inhibited and improved significantly by oral fucoidan supplementation with its antioxidant activity (Figure 6).

In conclusion, our current results indicate that fucoidan, with its potent antioxidant capacity, exerts cardio protective effects by reducing cellular ROS as well as its subsequent mitochondrial dysfunctions. Therefore, fucoidan could significantly ameliorate DOXO-induced cardiotoxicity by scavenging oxidative stress and preventing mitochondrial function injury. It is worth noting that fucoidan has been shown to inhibit a variety of cancers and can potentially be used in combination with DOXO to enhance tumor treatment while protecting cardiac function. We provide novel theoretical strategies for the DOXO-induced cardiotoxicity and explore a new idea for the treatment of cancer patients, such as dietary interventions.

## 4. Methods and Material

### 4.1. Animals

Healthy, male C57BL/6J mice of equal weight and seven weeks of age were offered by Beijing Vital River Laboratory Animal Technology Co., Ltd. (Beijing, China). The animals adapted to the laboratory conditions for no less than one week prior to the experiment. The animals were allowed to eat food and to drink water freely and were housed in a room with the temperature and the relative humidity maintained at 22 ± 2 °C and from 50% to 60%, respectively. All test procedures abided by the Guiding Principles for the Care and Use of Laboratory Animals and were reviewed and approved by the Committee on the Ethics of Animal Experiments of China Agricultural University (Beijing, China) (approval code: AW51102202-4-1; approval date: 15 January 2022).

### 4.2. Experimental Design

The mice were randomly divided into three groups, 6 mice in each group (*n* = 6). The experimental groups were (1) the control group, in which the mice were normally controlled with a standard diet (ctl group); (2) the doxorubicin group, in which the mice were treated with an equal volume of water for 14 days and then a cumulative dose of 7 mg/kg of DOXO (Cat. No. 23214-92-8, Targetmol, Waltham, MA, USA) for seven daily intraperitoneal injections (1 mg/kg/day, DOXO group), with no deaths associated with this administration regimen; and (3) the fucoidan plus DOXO group, in which the mice were orally treated with fucoidan from brown seaweed (Cat. No. F885876, Macklin, Shanghai, China) for 14 days at a dose of 500 mg/kg/day and then treated with the co-administration of oral fucoidan (1 h prior to DOXO) and injected with DOXO, with doses of DOXO being the same as that for the DOXO group (fucoidan + DOXO), while the other groups were injected an equivalent volume of 0.9% saline (Figure 1A).

### 4.3. Transthoracic Echocardiography

In vivo cardiac function was estimated by transthoracic echocardiography. The mice were anesthetized with chloral hydrate (Cat. No. C804536, Macklin, Shanghai, China) and placed in a supine position; then, the corresponding electrode was inserted into the upper right upper limp and the lower left lower limp. Echocardiography (ECG) was performed using a Vevo 3100 High-Resolution in Vivo Micro-Imaging System (FUJIFIUM Visual Sonics, Toronto, Ontario, Canada). The chests were shaved, and the left ventricle (LV) was tested in the parasternal short-axis and long-axis views at a frame rate of 233 Hz. The LV end-diastolic dimensions (LVEDD) and LV end-systolic dimensions (LVEDS) were used in the M-model images for an average of 3–5 heartbeats. End-diastolic and end-systolic are the phases corresponding to the T wave and R wave of the ECG. The left ventricular ejection fraction was calculated by the following formula: EF [%] = (LVIDd^3^ − LVIDs^3^)/LVIDd^3^ (EF, eject fraction; LVID, left ventricular diastolic; d, diastole; s, systole).

### 4.4. Measurement of Serum Biochemical Index

The mice were sacrificed by cervical dislocation after being anesthetized with chloral hydrate, and their hearts were excised and processed for further studies. Then, the serum of mice was collected by eyeball extraction and centrifuged at 4 °C, 4000 rpm for 20 min. The serum creatine kinase (CK), aspartate aminotransferase (AST) and lactate dehydrogenase (LDH-L) were measured by their respective kits (Cat. No. S03024, S03040, S03034, Rayto, Shenzhen, China). Creatine kinase isoenzyme (CK-MB) and serum lactate dehydrogenase isoenzyme 1 (LDH-1) were measured using their respective kits (Cat. No. C060, C058-e, Changchun Huili, China). Superoxide dismutase (SOD), malondialdehyde (MDA) and glutathione peroxidase (GSH-PX) were measured by their respective kits (Cat. No. A001-1, A003-1, CA005, Nanjing, China). All of the mentioned indicators were tested according to the manufacturer’s instructions.

### 4.5. Immunohistological Analysis

Hearts from each mouse were excised and washed in cold phosphate-buffered saline (Cat. No. D8537, Sigma-Aldrich, St. Louis, MO, USA, Merck & Co. Inc., Kenilworth, NJ, USA), then fixed overnight in 4% paraformaldehyde (pH 7.4) and embedded in paraffin. Cardiac fibrosis was measured by staining 5 μm-thick tissue sections serially, and the slices were dewaxed, rehydrated and subjected to hematoxylin-eosin (H&E) for histological examinations with an optical microscope. The cross-sectional cardio-myocyte area was analyzed by Image J software (version: 1.49, National Institutes of Health, Bethesda, MD, USA).

### 4.6. Interstitial Fibrosis Analysis

The sections were stained with the Picro Sirius Red Stain Kit (Cat. No. G1470, Beijing Solarbio Science, Beijing, China) to evaluate interstitial fibrosis. Heart slices were dewaxed, rehydrated and incubated with the Weigert ferrolin staining solution for 10–20 min, washed with distilled water, incubated with Picro Sirius Red for 1 h, then washed with water, dehydrated and mounted. ImageJ software was used to measure the positive staining (red) fibrotic area with the average percentage of fibrosis to the total area, which was calculated for six images of each heart.

### 4.7. Quantitative Real-Time PCR

The total RNA of the mice heart was separated and extracted using TRIzon Reagent (Cat. No. CW0580S, Cwbio, Beijing, China); then, the Nanodrop 2000 (Thermo Scientific, Waltham, MA, USA) was taken to measure the purity and concentration of the RNA. Using the HiScript III RT Super Mix (Cat. No. R323-01, Vazyme, Nanjing, China) for qPCR reverse transcription to obtain the cDNA; then, the expression quantities were measured using Taq Pro Universal SYBR qPCR Master Mix (Cat. No Q712-02, Vazyme, Nanjing, China). All of detective procedures followed the manufacturer’s instructions. Quantification was performed on the detective system and presented relative to *Gapdh*. Specificity was verified by the melting curve while each reaction was performed in triplicate, and the differences in gene expressions were calculated using the 2^−ΔΔCt^ method.

### 4.8. ATP and ROS Levels in Mice Heart

The enhanced ATP Assay Kit (Cat. No S0027, Beyotime, Shanghai, China) was used to detect ATP content in mice cardiac tissues. At first, the ATP working solution was added to the test wells to remove the background and then mixed with the tissue supernatant obtained from lysis for more than 2 s; finally, the RLU value was measured by a spectrophotometer. The efficiency of the test method can be referenced from the manual. The ROS content in the mice heart was evaluated using a fluorescent probe DCH-DA kit (Cat. No S0033S, Beyotime, Shanghai, China), and we conducted the experiments according to the manufacturer’s instructions. The heart tissue was lysed and added to DCFH-DA, incubated for 20 min in a constant temperature incubator at 37 °C and then cleaned with 1% sterile PBS three times. The obtained solution was added to a fluorescent microporous plate for quantitative analysis.

### 4.9. Statistical Analysis

Data in this study were presented as mean ± standard error of mean (SEM), and all statistical calculations were analyzed according GraphPad Prism 8 software (GraphPad Software, San Diego, CA, USA). The difference between two conditions were compared using an unpaired Student’s *t* test, and one-way analysis of variance (ANOVA) with Bonferroni correction was used to compare three or more conditions. Statistical significance was considered when the probability values (*p* values) were less than 0.05.

## Figures and Tables

**Figure 1 ijms-23-10685-f001:**
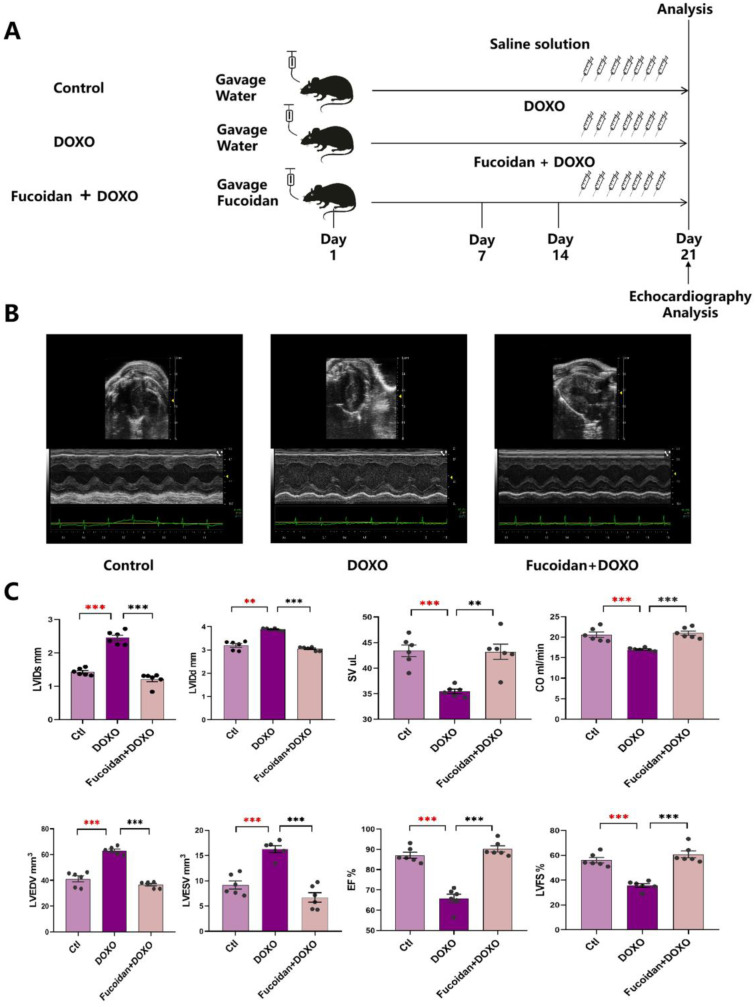
Fucoidan protects against DOXO-induced cardiotoxicity. (**A**) Schematic protocol of treatment and echocardiography in mice. C57/BL6J mice were divided into three groups randomly (*n* = 6 mice in each group). On day 0, mice in the pre-treated fucoidan + DOXO group were fed orally with fucoidan every day for 21 days, while the mice of the ctl and DOXO groups were given water gavage. On day 15, after daily pre-treatment with water or fucoidan, mice in the DOXO group and the pre-treated fucoidan + DOXO group were intraperitoneally injected with DOXO for the following 7 days, while mice of the ctl group were treated with the saline solution. Cardiac function was monitored by echocardiographic analysis on day 21. Mice were sacrificed after the analysis for subsequent ex vivo assays. (**B**) Fucoidan prevented DOXO-induced left ventricular dilation. Sample M-model short-axis echocardiographic image showing left ventricular dilation caused by DOXO and the protective effect of fucoidan in the mice of the fucoidan + DOXO group. (**C**) Left ventricular systole/diastole diameter (LVTDs/LVTDd) and left ventricular end-systolic/end-diastolic volume (LVESv/LVEDv) were significantly lower in fucoidan–DOXO-treated mice than in those in the DOXO group, and on the contrary, stroke volume (SV), cardiac output (CO), ejection fraction (EF) and left ventricular fractional shortening (LVFS) were significantly higher. ** *p* < 0.01; *** *p* < 0.001. The red asterisk is used to indicate the difference between the DOXO group and the Ctl group, and the black asterisk is used to indicate the difference between the DOXO group and the Fucoidan + DOXO group.

**Figure 2 ijms-23-10685-f002:**
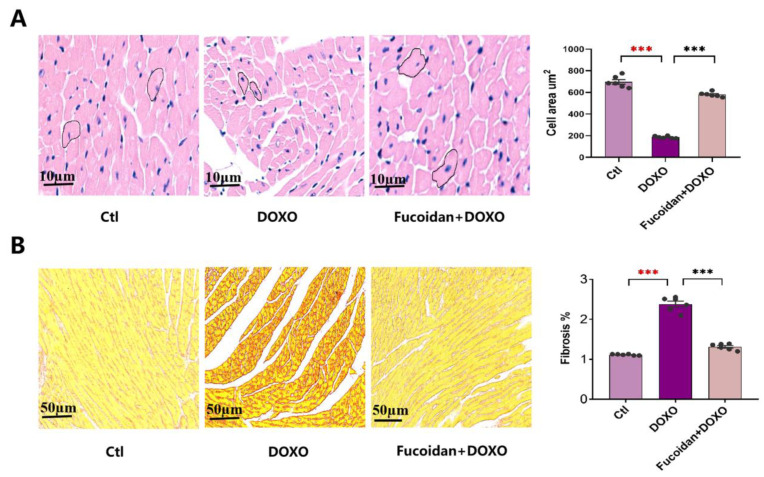
Effects on cardiac myocytes size and cardiac fibrosis of three groups, and fucoidan can improve DOXO-induced myocardial damage. (**A**) The cardiac myocardial cell area of ctl, DOXO and DOXO + fucoidan groups observed by H&E staining. (**B**) The rates of cardiac fibrosis of ctl, DOXO and DOXO + fucoidan groups observed by PicroSirius red staining. Each experiment was repeated in 6 samples. *** *p* < 0.001. The red asterisk is used to indicate the difference between the DOXO group and the Ctl group, and the black asterisk is used to indicate the difference between the DOXO group and the Fucoidan + DOXO group.

**Figure 3 ijms-23-10685-f003:**
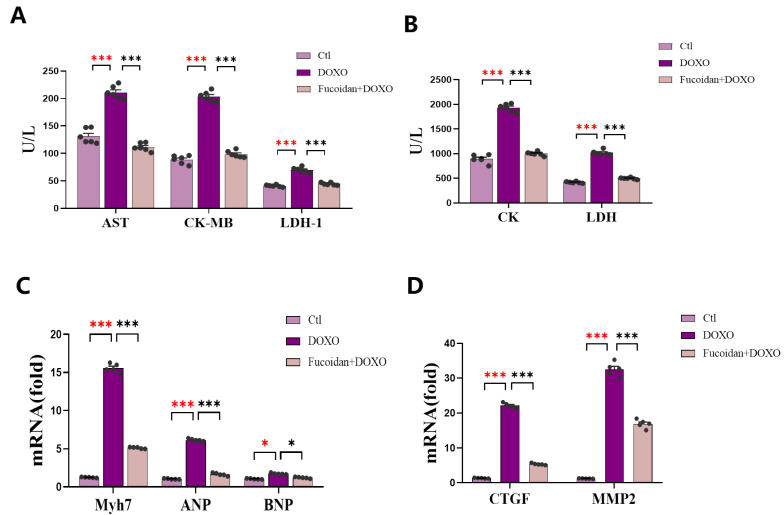
Fucoidan plays a key role in DOXO-induced cardiac structure and function damage prevention. (**A**) Levels of AST, CK-MB and LDH-1 in the mice serum. (**B**) Levels of CK and LDH in the mice serum. (**C**) Levels of cardiac dysfunction-related genes’ mRNA expression: *ANP*, *BNP* and *Myh7*. (**D**) Levels of cardiac remodeling-related genes mRNA expression: *CTGF* and *MMP2*. Repeating *n* = 6 in each group. Both C and D show that DOXO group had higher expression than ctl group, and fucoidan gavage was capable of impairing the cardiac function. * *p* < 0.05, *** *p* < 0.001. The red asterisk is used to indicate the difference between the DOXO group and the Ctl group, and the black asterisk is used to indicate the difference between the DOXO group and the Fucoidan + DOXO group.

**Figure 4 ijms-23-10685-f004:**
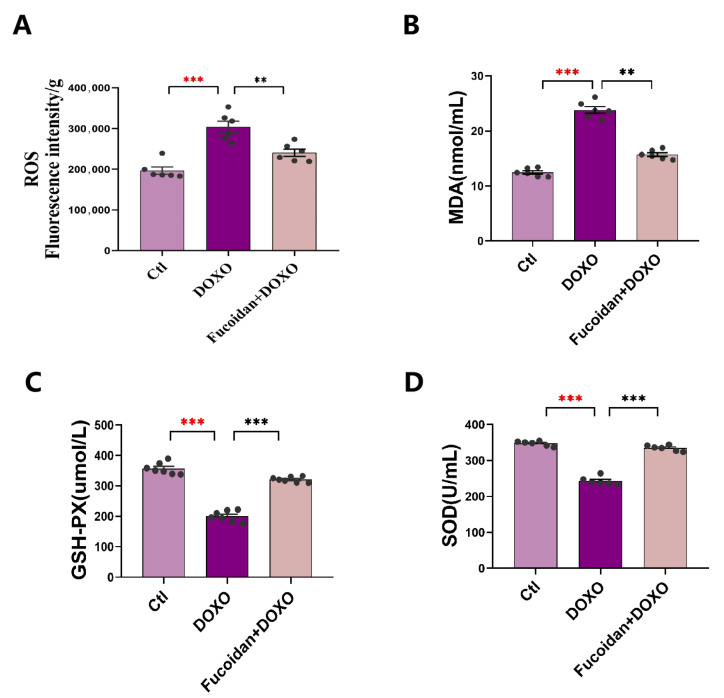
Fucoidan has a significant effect on reducing oxidative stress in DOXO-treated mice. (**A**) Levels of ROS in the mice heart tissues. (**B**) Levels of MDA in the mice serum. (**C**) Content of GSH-PX in the mice serum. (**D**) Content of SOD in the mice serum. *n* = 6 in each group. ** *p* < 0.05, *** *p* < 0.001. The red asterisk is used to indicate the difference between the DOXO group and the Ctl group, and the black asterisk is used to indicate the difference between the DOXO group and the Fucoidan + DOXO group.

**Figure 5 ijms-23-10685-f005:**
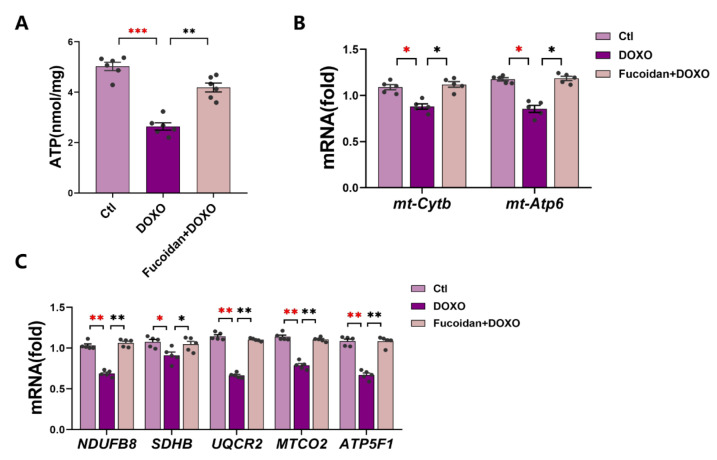
Fucoidan reduces DOXO-induced cardiotoxicity by preventing mitochondrial function injury. (**A**) Levels of ATP in the heart tissues of each mouse (all three groups). (**B**) Levels of mitochondrial function-related genes mRNA expression: *mt-Cytb* and *mt-Atp6*. (**C**) Levels of mitochondrial complex-related genes mRNA expression: *NDUFB8*, *SDHB*, *UQCR2*, *MTCO2* and *ATP5F1*. The results of (**B**,**C**) show that the DOXO group exhibits less expression than the ctl group while fucoidan administration can increase the expression level to improve the mitochondrial function. *n* = 6 in each group. * *p* < 0.05, ** *p* < 0.01, *** *p* < 0.001. The red asterisk is used to indicate the difference between the DOXO group and the Ctl group, and the black asterisk is used to indicate the difference between the DOXO group and the Fucoidan + DOXO group.

**Figure 6 ijms-23-10685-f006:**
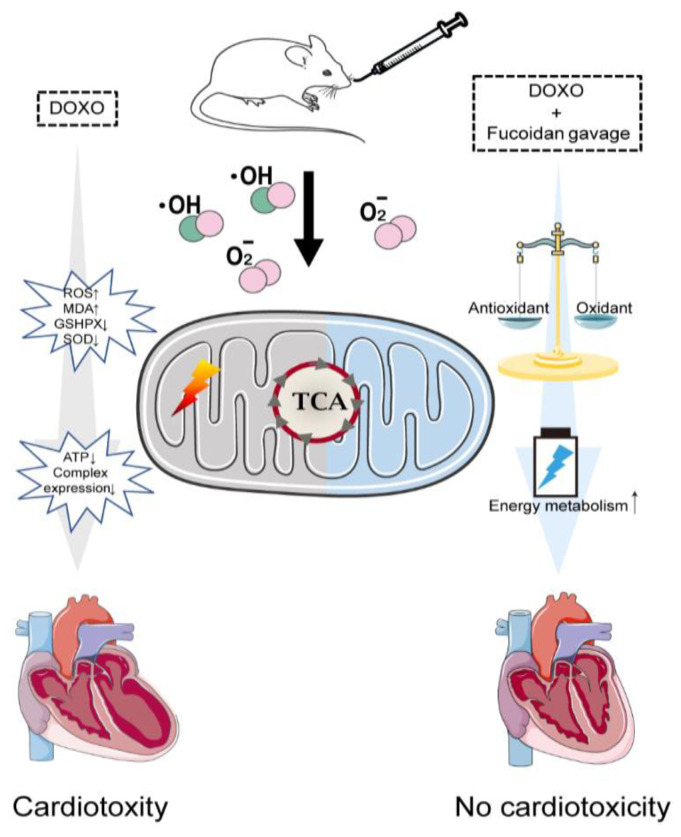
Schematic illustration of fucoidan preventing DOXO-induced cardiotoxicity. DOXO triggers cardiotoxicity via elevating oxidative stress and increasing mitochondrial function injury (**left**). Fucoidan protects against DOXO-induced cardiotoxicity by reducing oxidative stress and enhancing mitochondrial energy metabolism (**right**).

**Table 1 ijms-23-10685-t001:** Comparison of cardiac parameters between three treatments.

	Ctl (*n* = 6)	DOXO (*n* = 6)	Fucoidan + DOXO (*n* = 6)
	Mean ± SEM	Mean ± SEM	Mean ± SEM
LVID s (mm)	1.432 ± 0.043	2.456 ± 0.081 ***	1.210 ± 0.077 ***
LVID d (mm)	3.197 ± 0.076	3.882 ± 0.022 **	3.051 ± 0.035 ***
LVESV (mm^3^)	9.177 ± 0.817	16.279 ± 0.660 ***	6.716 ± 0.934 ***
LVEDV (mm^3^)	41.128 ± 2.323	63.031 ± 1.184 ***	36.582 ± 0.996 ***
Stroke Volume (L)	43.426 ± 1.135	35.480 ± 0.421 ***	43.207 ± 1.489 ***
Ejection Fraction (%)	87.164 ± 1.521	65.868 ± 2.086 ***	90.287 ± 1.504 ***
Fractional Shortening (%)	56.107 ± 2.116	35.684 ± 1.504 ***	60.718 ± 2.797 ***
Cardiac Output (mL/min)	20.586 ± 0.670	17.031 ± 0.149 ***	21.003 ± 0.524 ***
LV Mass (mg)	25.916 ± 0.404	26.001 ± 1.354	29.988 ± 1.531
LVAW s (mm)	0.307 ± 0.019	0.212 ± 0.021 **	0.302 ± 0.009 **
LVAW d (mm)	0.341 ± 0.007	0.258 ± 0.026 *	0.326 ± 0.019
LVPW s (mm)	0.350 ± 0.021	0.252 ± 0.013 **	0.330 ± 0.017 **
LVPW d (mm)	0.371 ± 0.008	0.299 ± 0.009 ***	0.431 ± 0.014 ***

LVID, left ventricular diameter; LVESV, left ventricular end-systolic volume; LVEDV, left ventricular end-diastolic volume; SV, stroke volume; EF, ejection fraction; FS, fractional shortening; CO, cardiac output; LV Mass, left ventricular mass; LVAW, left ventricular anterior wall thickness; LVPW, left ventricular posterior wall thickness; s, systole; d, diastole. * *p* < 0.05, ** *p* < 0.01, *** *p* < 0.001. The red asterisk is used to indicate the difference between the DOXO group and the Ctl group, and the black asterisk is used to indicate the difference between the DOXO group and the Fucoidan + DOXO group.

## Data Availability

Not applicable.

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
