# Peer review of "Fucoidan Protects against Doxorubicin-Induced Cardiotoxicity by Reducing Oxidative Stress and Preventing Mitochondrial Function Injury"

_ijms, 2022, doi:10.3390/ijms231810685_

Round 1

Reviewer 1 Report

The authors found that fucoidan is protective against DOXO-induced myocardial toxicity by improving mitochondrial function. Cardiotoxicity caused by DOXO is a major topic of concern for many patients, and this study is significant not only because it demonstrates the efficacy of the therapeutic agent, but also because it demonstrates the mechanism. This article is also interesting as a study linking the compound fucoidan to heart failure. To further clarify the focus of this article, I would ask the following experiment.

<main>

・Regarding the figure2 data.

The image in Figure A appears to be a low magnification image only for DOXO. Please double-check the scale and resolution of the image.

The Sirius red-stained image in Figure B appears to be a background or artifact. I don't think that HE staining, at least with figure A, shows so much significant fibrosis. It is recommended to normalize the overall red color with cytoplasmic staining or Masson trichrome staining.

Despite the same scale in Figures A and B, the cell sizes are too different. Please check the details of the images.

・In figure 5, It is risky to suggest mitochondrial biosynthesis based solely on the results of elevated mRNA expression. Mitochondrial protein and mRNA levels are often not linked in the pathogenesis of cardiomyopathy. If experimental techniques are available, it is recommended that the amount of mitochondrial respiratory chain complexes be analyzed by Western blotting. A cocktail of antibodies is also available.

・The three points of reduced mitochondrial ROS, increased antioxidant capacity, and increased mitochondrial biosynthesis seem to be an independent story. These are difficult to summarize in writing, so it is worth considering how these phenomena occur and how they contribute to cardioprotection, at least in terms of fucoidan’s action.

<minor>

・Regarding the "Fucoidan attenuates DOXO-induced myocardial atrophy and cardiac fibrosis" part of 3.2. I feel the meaning of examining the size of the cardiomyocytes is ambiguous. When considering heart failure, myocardial cells are basically hypertrophied, but there are many reports of cell atrophy in DOXO. To avoid misleading many cardiac researchers, it would be desirable to introduce at the beginning of the text that DOXO often causes cell atrophy or the mechanism of cell atrophy caused by DOXO.

Author Response

# Reviewer 1

The authors found that fucoidan is protective against DOXO-induced myocardial toxicity by improving mitochondrial function. Cardiotoxicity caused by DOXO is a major topic of concern for many patients, and this study is significant not only because it demonstrates the efficacy of the therapeutic agent, but also because it demonstrates the mechanism. This article is also interesting as a study linking the compound fucoidan to heart failure. To further clarify the focus of this article, I would ask the following experiment.

RESPONSE: We sincerely thank the reviewer for the encouraging and insightful comments.

<main>

・Regarding the figure2 data.

The image in Figure A appears to be a low magnification image only for DOXO. Please double-check the scale and resolution of the image.

RESPONSE: We thank the reviewer for this helpful comment. We have double-checked and confirmed that all the images of the three groups were taken under the same magnification and resolution. Actually, the apparent low magnification of the image is due to the pathological changes of cardiomyocyte shrinkage upon DOXO-induced cardiac injury.

The Sirius red-stained image in Figure B appears to be a background or artifact. I don't think that HE staining, at least with figure A, shows so much significant fibrosis. It is recommended to normalize the overall red color with cytoplasmic staining or Masson trichrome staining.

RESPONSE: We thank the reviewer for pointing out this important issue. We agree with the reviewer that there is no obvious fibrosis in HE staining (Figure A). As shown in the results section, HE staining was used to indicate the overall pathology and the size of myocardial cells. In contrast, the Sirius red staining was used to evaluate the organization and accumulation of collagen fibers in cardiac tissues. Sirius red-stain is a strong anionic acid dye that readily reacts with basic groups in collagen fibers. Collagen fibers can be specifically stained scarlet. It is a widely used method to discriminate cardiac tissue fibrosis. The type I collagen fibers of the myocardium can be observed to be stained red using polarized light microscopy, and the rest of the myocardial tissue is stained yellow. The staining pattern and background in our results is consistent with that in the other literatures [1, 2]. Therefore, we confirm that the Sirius red-stained image in Figure B is not a background or artifact. The enhanced red signal in the DOXO group is indicative of an elevation of fibrosis compared with either control or DOXO + fucoidan group.

  1.  Brakenhielm, E.; González, A.; Díez, J., Role of Cardiac Lymphatics in Myocardial Edema and Fibrosis: JACC Review Topic of the Week. J Am Coll Cardiol 2020, 76, (6), 735-744.
  2.  Russo, M.; Guida, F.; Paparo, L.; Trinchese, G.; Aitoro, R.; Avagliano, C.; Fiordelisi, A.; Napolitano, F.; Mercurio, V.; Sala, V.; Li, M.; Sorriento, D.; Ciccarelli, M.; Ghigo, A.; Hirsch, E.; Bianco, R.; Iaccarino, G.; Abete, P.; Bonaduce, D.; Calignano, A.; Berni Canani, R.; Tocchetti, C. G., The novel butyrate derivative phenylalanine-butyramide protects from doxorubicin-induced cardiotoxicity. Eur J Heart Fail 2019, 21, (4), 519-528.

Despite the same scale in Figures A and B, the cell sizes are too different. Please check the details of the images.

RESPONSE: We sincerely thank the reviewer for pointing out this error. We agree with the reviewer, and re-labeled the scale bar in the revised Figure B (see page 7).

・In figure 5, It is risky to suggest mitochondrial biosynthesis based solely on the results of elevated mRNA expression. Mitochondrial protein and mRNA levels are often not linked in the pathogenesis of cardiomyopathy. If experimental techniques are available, it is recommended that the amount of mitochondrial respiratory chain complexes be analyzed by Western blotting. A cocktail of antibodies is also available.

RESPONSE: We totally agree with the reviewer that it is indeed risky to suggest mitochondrial biogenesis based solely on the results of elevated mRNA expression. Indeed, we observed that doxorubicin caused dramatic decrease of ATP production in cardiac tissues. The downregulation of the gene expression of mitochondrial oxidative respiratory chain explains the reduction of ATP, but not mitochondrial biosynthesis. In order to avoid this confusion, we rephrased the related sentences to make it clear (see page 9). In addition, we also thank the reviewer for the valuable suggestion to measure the respiratory chain complexes on the protein level. As suggested, we have searched the antibodies. However, the delivery date of all the antibodies exceeds 10 days or even longer times, making it impossible to revise back the manuscript within due time.

・The three points of reduced mitochondrial ROS, increased antioxidant capacity, and increased mitochondrial biosynthesis seem to be an independent story. These are difficult to summarize in writing, so it is worth considering how these phenomena occur and how they contribute to cardio protection, at least in terms of fucoidan’s action.

RESPONSE: We thank the reviewer for this constructive comment. As we have mentioned in the Introduction section, the main mechanism of DOXO in inducing cardiac pathology is through producing a large amount of reactive oxygen species (ROS), which in turn leads to mitochondrial dysfunction and cellular injury (see page 2). Therefore, fucoidan, with its potent antioxidant capacity, exerts the cardio protective effects by reducing cellular ROS as well as the subsequent mitochondrial dysfunctions. As suggested, we have rephrased the related sentences to make it clear (see page 11).

<minor>

・Regarding the "Fucoidan attenuates DOXO-induced myocardial atrophy and cardiac fibrosis" part of 3.2. I feel the meaning of examining the size of the cardiomyocytes is ambiguous. When considering heart failure, myocardial cells are basically hypertrophied, but there are many reports of cell atrophy in DOXO. To avoid misleading many cardiac researchers, it would be desirable to introduce at the beginning of the text that DOXO often causes cell atrophy or the mechanism of cell atrophy caused by DOXO.

RESPONSE: We sincerely thank the reviewer for the helpful comment. We agree that cardiomyocytes are basically hypertrophied in heart failure, while DOXO often causes cell atrophy. As suggested, we have added additional text and two references into the Introduction section to make it clear (see page 1).

Reviewer 2 Report

In this manuscript Ji et al evaluate the protective effect of fucoidan on doxorubicin-induced cardiotoxicity. The authors conclude that fucoidan reduced oxidative stress and prevents mitochondrial function injury induced by doxorubicin.

Critical Comments

1)      Does Fucoidan reduce the antitumour activity of doxorrubucin?

2)      Treated controls with Fucoidan are absent in this work.

Author Response

# Reviewer 2

In this manuscript Ji et al evaluate the protective effect of fucoidan on doxorubicin-induced cardiotoxicity. The authors conclude that fucoidan reduced oxidative stress and prevents mitochondrial function injury induced by doxorubicin.

RESPONSE: We sincerely thank the reviewer for the careful and critical review of our manuscript.

Critical Comments

  • Does Fucoidan reduce the antitumour activity of doxorrubucin?

RESPONSE: Thank you for this interesting comment. Fucoidan, as a natural active sulfate polysaccharide, exerts a wide range of biological activities, such as anti-tumor, antiviral, antioxidation, and anti-inflammation. In fact, several studies have shown that fucoidan is resistant to a variety of cancers, and fucoidan can be combined with clinical drugs as an adjuvant for anticancer effect [1-6]. Doxorubicin is most commonly used in patients with breast cancer. Notably, fucoidan can treat breast cancer by improving gut microbiota, and the combination of fucoidan and doxorubicin will achieve better therapeutic effect on breast cancer patients [7]. Therefore, there is currently no clear evidence that fucoidan could reduce the antitumor activity of doxorubicin.

  1. 1.  Hsu, H. Y.; Hwang, P. A., Clinical applications of fucoidan in translational medicine for adjuvant cancer therapy. Clin Transl Med 2019, 8, (1), 15.
  2. 2.  Chantree, P.; Na-Bangchang, K.; Martviset, P., Anticancer Activity of Fucoidan via Apoptosis and Cell Cycle Arrest on Cholangiocarcinoma Cell. Asian Pac J Cancer Prev 2021, 22, (1), 209-217.
  3. 3.  Kwak, J. Y., Fucoidan as a marine anticancer agent in preclinical development. Mar Drugs 2014, 12, (2), 851-70.
  4. 4.  Narayani, S. S.; Saravanan, S.; Ravindran, J.; Ramasamy, M. S.; Chitra, J., In vitro anticancer activity of fucoidan extracted from Sargassum cinereum against Caco-2 cells. Int J Biol Macromol 2019, 138, 618-628.
  5. 5.  P, A.; K, A.; L, S.; M, M.; K, M., Anticancer effect of fucoidan on cell proliferation, cell cycle progression, genetic damage and apoptotic cell death in HepG2 cancer cells. Toxicol Rep 2019, 6, 556-563.
  6. 6.  Senthilkumar, K.; Kim, S. K., Anticancer effects of fucoidan. Adv Food Nutr Res 2014, 72, 195-213.
  7. 7.  Xue, M.; Ji, X.; Liang, H.; Liu, Y.; Wang, B.; Sun, L.; Li, W., The effect of fucoidan on intestinal flora and intestinal barrier function in rats with breast cancer. Food Funct 2018, 9, (2), 1214-1223.

  • Treated controls with Fucoidan are absent in this work.

RESPONSE: We thank the reviewer for pointing out this issue. Based on the three experimental groups, i.e., the control group, DOXO group, and Fucoidan+DOXO group, we are able to clearly demonstrate the protective effects of fucoidan on doxorubicin-induced cardiotoxicity. Therefore, the treated control with fucoidan is unnecessary.

Round 2

Reviewer 1 Report

Authors addressed experimentally all the concerns raised by this reviewer. The manuscript is now a strong candidate for publication.

Author Response

Authors addressed experimentally all the concerns raised by this reviewer. The manuscript is now a strong candidate for publication.

RESPONSE: We sincerely thank the reviewer for the encouraging and insightful comments.

Reviewer 2 Report

The control group treated with fucoidan is necessary. Please see similar article:

https://bpspubs.onlinelibrary.wiley.com/doi/10.1111/bph.12795

https://www.nmcd-journal.com/article/S0939-4753%2817%2930037-6/fulltext

Author Response

The control group treated with fucoidan is necessary. Please see similar article:

https://bpspubs.onlinelibrary.wiley.com/doi/10.1111/bph.12795

https://www.nmcd-journal.com/article/S0939-4753%2817%2930037-6/fulltext.

RESPONSE: We sincerely thank the reviewer for the constructive comment. Indeed, the first article used quercetin (QCT) alone, and no significant difference was found between QCT alone group and the control group for all parameters tested by the authors. For the second article, the authors set up control groups in the cell experiments, however, there was no specific control group in the mouse experiments.

The control group treated with fucoidan alone may be used to detect toxic side effects of fucoidan, but currently it is believed that fucoidan is a natural polysaccharide derived from food, and the established safety and availability of fucoidan as a food supplement or dietary supplement have been verified (https://www.ncbi.nlm.nih.gov/pmc/articles/PMC3210604/; https://ift.onlinelibrary.wiley.com/doi/10.1111/j.1750-3841.2012.02966.x). More importantly, the toxic side effects of fucoidan are not the focus of this manuscript.

In fact, consistent with our study design, many published articles used the same grouping strategy in animal experiments. Please see the list of examples as follows.

  1. Liu, Y.; Asnani, A.; Zou, L.; Bentley, V. L.; Yu, M.; Wang, Y.; Dellaire, G.; Sarkar, K. S.; Dai, M.; Chen, H. H.; Sosnovik, D. E.; Shin, J. T.; Haber, D. A.; Berman, J. N.; Chao, W.; Peterson, R. T., Visnagin protects against doxorubicin-induced cardiomyopathy through modulation of mitochondrial malate dehydrogenase. Science translational medicine 2014, 6, (266), 266ra170.
  2. Li, W.; Wang, X.; Liu, T.; Zhang, Q.; Cao, J.; Jiang, Y.; Sun, Q.; Li, C.; Wang, W.; Wang, Y., Harpagoside Protects Against Doxorubicin-Induced Cardiotoxicity via P53-Parkin-Mediated Mitophagy. Frontiers in cell and developmental biology 2022, 10, 813370.
  3. Lerida-Viso, A.; Estepa-Fernandez, A.; Morella-Aucejo, A.; Lozano-Torres, B.; Alfonso, M.; Blandez, J. F.; Bisbal, V.; Sepulveda, P.; Garcia-Fernandez, A.; Orzaez, M.; Martinez-Manez, R., Pharmacological senolysis reduces doxorubicin-induced cardiotoxicity and improves cardiac function in mice. Pharmacological research 2022, 183, 106356.
  4. Kim, B. S.; Park, I. H.; Lee, A. H.; Kim, H. J.; Lim, Y. H.; Shin, J. H., Sacubitril/valsartan reduces endoplasmic reticulum stress in a rat model of doxorubicin-induced cardiotoxicity. Archives of toxicology 2022, 96, (4), 1065-1074.
  5. Qin, Y.; Lv, C.; Zhang, X.; Ruan, W.; Xu, X.; Chen, C.; Ji, X.; Lu, L.; Guo, X., Neuraminidase1 Inhibitor Protects Against Doxorubicin-Induced Cardiotoxicity via Suppressing Drp1-Dependent Mitophagy. Frontiers in cell and developmental biology 2021, 9, 802502.
  6. Lu, J.; Li, J.; Hu, Y.; Guo, Z.; Sun, D.; Wang, P.; Guo, K.; Duan, D. D.; Gao, S.; Jiang, J.; Wang, J.; Liu, P., Chrysophanol protects against doxorubicin-induced cardiotoxicity by suppressing cellular PARylation. Acta pharmaceutica Sinica. B 2019, 9, (4), 782-793.
  7. Liu, D.; Ma, Z.; Xu, L.; Zhang, X.; Qiao, S.; Yuan, J., PGC1alpha activation by pterostilbene ameliorates acute doxorubicin cardiotoxicity by reducing oxidative stress via enhancing AMPK and SIRT1 cascades. Aging 2019, 11, (22), 10061-10073.
  8. Hiona, A.; Lee, A. S.; Nagendran, J.; Xie, X.; Connolly, A. J.; Robbins, R. C.; Wu, J. C., Pretreatment with angiotensin-converting enzyme inhibitor improves doxorubicin-induced cardiomyopathy via preservation of mitochondrial function. The Journal of thoracic and cardiovascular surgery 2011, 142, (2), 396-403 e3.
  9. Riad, A.; Bien, S.; Westermann, D.; Becher, P. M.; Loya, K.; Landmesser, U.; Kroemer, H. K.; Schultheiss, H. P.; Tschope, C., Pretreatment with statin attenuates the cardiotoxicity of Doxorubicin in mice. Cancer research 2009, 69, (2), 695-9.
  10. Ibrahim, M. A.; Ashour, O. M.; Ibrahim, Y. F.; El-Bitar, H. I.; Gomaa, W.; Abdel-Rahim, S. R., Angiotensin-converting enzyme inhibition and angiotensin AT(1)-receptor antagonism equally improve doxorubicin-induced cardiotoxicity and nephrotoxicity. Pharmacological research 2009, 60, (5), 373-81.
  11. Lebrecht, D.; Geist, A.; Ketelsen, U. P.; Haberstroh, J.; Setzer, B.; Walker, U. A., Dexrazoxane prevents doxorubicin-induced long-term cardiotoxicity and protects myocardial mitochondria from genetic and functional lesions in rats. British journal of pharmacology 2007, 151, (6), 771-8.
  12. Yagmurca, M.; Fadillioglu, E.; Erdogan, H.; Ucar, M.; Sogut, S.; Irmak, M. K., Erdosteine prevents doxorubicin-induced cardiotoxicity in rats. Pharmacological research 2003, 48, (4), 377-82.

Therefore, we insist that the current experimental design (the control group, DOXO group, and fucoidan+DOXO group) is reasonable and sufficient to demonstrate the protective effect of fucoidan against doxorubicin-induced cardiotoxicity, and the control group treated with fucoidan alone is not necessary.